# No-Regret Learning and Mixed Nash Equilibria:
# They Do Not Mix

**Lampros Flokas**\*
Department of Computer Science
Columbia University
New York, NY 10025
lamflokas@cs.columbia.edu

**Emmanouil V. Vlatakis-Gkaragkounis**\*
Department of Computer Science
Columbia University
New York, NY 10025
emvlatakis@cs.columbia.edu

**Thanasis Lianeas**\*
School of Electrical and Computer Engineering
National Technical University of Athens
Athens,Greece
lianeas@corelab.ntua.gr

**Panayotis Mertikopoulos**
Univ. Grenoble Alpes, CNRS, Inria, LIG &
Criteo AI Lab
panayotis.mertikopoulos@imag.fr

**Georgios Piliouras**
Engineering Systems and Design
Singapore University of Technology and Design
Singapore
georgios@sutd.edu.sg

## Abstract

Understanding the behavior of no-regret dynamics in general $N$-player games is a fundamental question in online learning and game theory. A folk result in the field states that, in finite games, the empirical frequency of play under no-regret learning converges to the game's set of coarse correlated equilibria. By contrast, our understanding of how the day-to-day behavior of the dynamics correlates to the game's *Nash* equilibria is much more limited, and only *partial* results are known for *certain* classes of games (such as zero-sum or congestion games). In this paper, we study the dynamics of *follow the regularized leader* (FTRL), arguably the most well-studied class of no-regret dynamics, and we establish a sweeping negative result showing that *the notion of mixed Nash equilibrium is antithetical to no-regret learning*. Specifically, we show that any Nash equilibrium which is not *strict* (in that every player has a unique best response) cannot be stable and attracting under the dynamics of FTRL. This result has significant implications for predicting the outcome of a learning process as it shows unequivocally that only strict (and hence, *pure*) Nash equilibria can emerge as stable limit points thereof.

## 1 Introduction

Regret minimization is one of the most fundamental requirements for online learning and decision-making in the presence of uncertainty and unpredictability [11]. Defined as the difference between the cumulative performance of an adaptive policy and that of the best fixed action in hindsight, the regret of an agent provides a concise and meaningful benchmark for quantifying the ability of an online algorithm to adapt to an otherwise unknown and unpredictable environment.

---

Arguably, the most widely studied class of no-regret algorithms is the general algorithmic scheme known as *follow the regularized leader* (FTRL) [56, 57]. This umbrella learning framework includes as special cases the multiplicative weights update (MWU) [2, 3, 32, 62] and online gradient descent (OGD) algorithms [64], both of which achieve a min-max optimal $\mathcal{O}(T^{1/2})$ regret guarantee. For obvious reasons, the ability of FTRL to adapt optimally to an unpredictable environment makes them ideal for applying them in multi-agent environments – i.e., games. In this case, if all agents adhere to a no-regret learning process based on FTRL (or one of its variants), as the sequence of play becomes more predictable, stronger regret guarantees are achievable, possibly down to constant regret, see e.g., [5, 6, 20, 30, 37, 38, 50, 58] and references therein. As such, several crucial questions arise:

What are the *game-theoretic implications* of the no-regret guarantees of FTRL?
*Do the dynamics of FTRL converge to an equilibrium of the underlying game?*

A folk answer to this question is that "*no-regret learning converges to equilibrium in all games*" [43], suggesting in this way that no-regret dynamics inherently gravitate towards game-theoretically meaningful states. However, at this level of abstraction, both the *type of convergence* as well as the specific *notion of equilibrium* that go in this statement are not as strong as one would have hoped for. Formally, the only precise conclusion that can be drawn is as follows: *under a no-regret learning procedure, the empirical frequency of play converges to the game's set of coarse correlated equilibria* [23, 24].

This leads to an important disconnect with standard game-theoretic solution concepts on several grounds. First, even in 2-player games, coarse correlated equilibria may be exclusively supported on *strictly* dominated strategies [60], so they fail even the most basic requirements of rationalizability [19, 22]. Second, the archetypal game-theoretic solution concept is that of *Nash equilibrium* (NE), and convergence to a Nash equilibrium is a much more tenuous affair: since no-regret dynamics are, by construction, uncoupled (in the sense that a player's update rule does not *explicitly* depend on the payoffs of other players), the impossibility result of Hart & Mas-Colell [25] precludes the convergence of no-regret learning to Nash equilibrium in *all* games. This is consistent with the numerous negative complexity results for finding a Nash equilibrium [18, 54]: an incremental method like FTRL simply cannot have enough power to overcome PPAD completeness and converge to Nash equilibrium given adversarially chosen initial conditions.

In view of the above, a natural test of whether the dynamics of FTRL favor convergence to a Nash equilibrium is to see whether they eventually stabilize and converge to it when initialized nearby. In more precise language, *are Nash equilibria asymptotically stable in the dynamics of FTRL?* And, perhaps more importantly, *are all Nash equilibria created equal in this regard?*

**Our contributions.**  We establish a stark and robust dichotomy between how the dynamics of FTRL treat Nash equilibria in *mixed* (i.e., randomized) vs. *pure* strategies. For the case of mixed Nash equilibria we establish a sweeping negative result to the effect that *the notion of mixed Nash equilibrium is antithetical to no-regret learning*. More precisely, we show that any Nash equilibrium which is not *strict* (in the sense that every player has a unique best response) cannot be stable and attracting under the dynamics of FTRL. Schematically:

**Informal Theorem:** Asymptotically stable point for FTRL $\implies$ Pure Nash equilibrium

**Equivalently:** Mixed Nash equilibrium $\implies$ Not asymptotically stable under FTRL

The linchpin of our analysis is the following striking property of the FTRL dynamics: when viewed in the space of "payoffs" (their natural state space), *they preserve volume irrespective of the underlying game.* More precisely, the Lebesgue measure of any open set of initial conditions in the space of payoffs remains invariant as it is carried along the flow of the FTRL dynamics (cf. Fig. 2). Importantly, this result is *not* true in the problem's "primal" space, i.e., the space of the player's mixed strategies: here, sets of initial conditions can expand or contract indefinitely under the standard Euclidean volume form.

This duality between payoffs and strategies is the leitmotif of our approach and has a number of important consequences. First, exploiting the volume-preservation property of FTRL, we show that no interior Nash equilibrium (and, furthermore, no closed set in the interior of the strategy space) can be asymptotically stable under the dynamics of FTRL, as this effectively would necessitate volume contraction in the interior of the space (Theorem 4.2).

To move beyond this result and disqualify *all* non-strict Nash equilibria (not just interior ones) more intricate arguments are required. In this case, a fundamental distinction arises between classes of dynamics that may attain the boundary of the players' strategy space in finite time versus those that do not. The first case concerns FTRL dynamics with an everywhere-differentiable regularizer, like the Euclidean regularizer that gives rise to OGD and the associated projection dynamics. The second concerns dynamics where the regularizer becomes *steep* at the boundary of the strategy simplex, e.g., like the Shannon-Gibbs entropy that gives rise to the multiplicative weights update (MWU) algorithm and the replicator dynamics. While the interior of the strategy simplex is invariant for the second class of dynamics, this is not the case for the former: in Euclidean-like cases, the support of the mixed strategy of an agent may change over time. This leads to an essential dichotomy in the boundary behavior of different classes of FTRL dynamics. Nonetheless, despite the qualitatively distinct long-run behavior of the dynamics, a unified message emerges: *under the dynamics of FTRL, only strict Nash equilibria survive* (Theorem 4.3).

Finally, for the case of steep, entropy-like regularizers we prove that not only their asymptotically stable points but much more generally *any* asypmptotically stable set must contain at least one pure strategy profile (Theorem 4.5).

**Related work.** The regret properties of FTRL have given rise to a vast corpus of literature which we cannot hope to review here; for an appetizer, we refer the reader to [10, 56] and references therein. On the other hand, the long-run behavior of FTRL in games (even finite ones) is nowhere near as well understood. A notable exception to this is the case of the replicator dynamics which have been studied extensively due to their origins and connection with evolutionary game theory, cf. [26, 55, 59, 63] for a review. For the replicator dynamics, a special instance of the volume preservation principle was first discovered by Akin [1] and ultimately gave rise to the so-called "folk theorem" of evolutionary game theory:[2] in population games, the notions of strict Nash equilibrium and asymptotic stability coincide [27]. This instability of mixed Nash equilibria plays a major role in the theory of population games as it shows that even the weakest form of mixing cannot be stable in an evolutionary sense. The volume preservation result that we establish here can be seen as a much more general "learning analogue" of this biological principle and provides an important link between population dynamics and the theory of online learning in games.

Recent work has examined the non-convergence of FTRL dynamics in more specialized settings. Coucheney et al. [17] established a version of the folk theorem of evolutionary game theory for a subclass of "decomposable", steep FTRL dynamics. By contrast, Mertikopoulos et al. [38] focused on two-player *zero-sum* games (and networked versions thereof), and showed that almost all trajectories of FTRL orbit interior equilibria at a fixed distance without ever converging to equilibrium, generalizing the previous analysis for replicator dynamics by Piliouras & Shamma [49]. This is an interior equilibrium avoidance result, but one that *uniquely concerns zero-sum games*. Although the above results apply for continuous-time dynamics, in discrete-time non-convergence results only become stronger. Bailey & Piliouras [4] proved that discrete-time FTRL diverges away from the Nash equilibrium in zero-sum games, whereas Cheung & Piliouras [12] established Lyapunov chaos (volume-expansion, butterfly effects). Understanding the detailed geometry of non-equilibrating FTRL dynamics, e.g., periodicity/chaos, is an interesting direction where volume analysis has found application [6, 8, 13, 39, 40, 48]. Non-convergence, recurrence results have recently been established for FTRL dynamics via volume analysis even outside normal form games, e.g., in non-convex non-concave min-max differential games [61] and imperfect information zero-sum games [47]. Finally, such instability, non-convergence results have inspired new, dynamics-based, solution concepts for games that generalize strict Nash while allowing cyclic, recurrent behavior [28, 44–46, 53].

In the converse direction, a complementary research thread has shown strict Nash equilibria are asymptotically stable under several incarnations of the FTRL dynamics [9, 15, 17, 34–37]. Our paper establishes the *converse* to this stability result, thus leading to the the following overarching principle (which covers all generic $N$-player games):

$$\textit{Asymptotic stability under FTRL} \iff \textit{Strict Nash equilibrium}$$

This result has significant implications for predicting the outcome of a learning process as it shows unequivocally that its pointwise stable outcomes are *precisely* the strict (and hence, *pure*) Nash equilibria of the underlying game.

## 2 Preliminaries

**Notation.** If $f$ is a function of a single variable, we will abuse notation slightly and extend it to vector variables $x \in \mathbb{R}^n$ by letting $f(x) \leftarrow (f(x_1), \ldots, f(x_n))$. We will also understand inequalities involving vectors component-wise, i.e., $(x_1, \ldots, x_n) > 0$ means that $x_i > 0$ for all $i = 1, \ldots, n$.

**The game.** Throughout the sequel, we will focus on finite games. Formally, a *finite game in normal form* is defined as a tuple $\Gamma \equiv \Gamma(\mathcal{N}, \mathcal{A}, u)$ consisting of (*i*) a finite set of *players* $i \in \mathcal{N} = \{1, \ldots, N\}$; (*ii*) a finite set of *actions* (or *pure strategies*) $\mathcal{A}_i = \{\alpha_1, \ldots, \alpha_{n_i}\}$ per player $i \in \mathcal{N}$; and (*iii*) each player's payoff function $u_i \colon \mathcal{A} \to \mathbb{R}$, where $\mathcal{A} := \prod_i \mathcal{A}_i$ denotes the ensemble of all possible *action profiles* $\alpha = (\alpha_1, \ldots, \alpha_N)$. In this general context, players can also play *mixed strategies*, i.e., probability distributions $x_i = (x_{i\alpha_i})_{\alpha_i \in \mathcal{A}_i} \in \Delta(\mathcal{A}_i)$ over their pure strategies $\alpha_i \in \mathcal{A}_i$. Collectively, we will write $\mathcal{X}_i := \Delta(\mathcal{A}_i)$ for the mixed strategy space of player $i$ and $\mathcal{X} := \prod_i \mathcal{X}_i$ for the space of all mixed strategy profiles $x = (x_1, \ldots, x_N)$.

Given a mixed profile $x \in \mathcal{X}$, the corresponding expected payoff of player $i$ will be

$$u_i(x) = \sum_{\alpha_1 \in \mathcal{A}_1} \cdots \sum_{\alpha_N \in \mathcal{A}_N} x_{1,\alpha_1} \cdots x_{N,\alpha_N} u_i(\alpha_1, \ldots, \alpha_N). \tag{1}$$

To keep track of the payoffs of each individual action, we will also write

$$v_{i\alpha_i}(x) := u_i(\alpha_i; x_{-i}) \tag{2}$$

for the payoff of the pure strategy $\alpha_i \in \mathcal{A}_i$ in the mixed profile $x = (x_i; x_{-i}) \in \mathcal{X}$.[3] Hence, writing $v_i(x) := (v_{i\alpha_i}(x))_{\alpha_i \in \mathcal{A}_i} \in \mathbb{R}^{\mathcal{A}_i}$ for the *payoff vector* of player $i$, we get the compact expression

$$u_i(x) = \langle v_i(x), x_i \rangle = \sum_{\alpha_i \in \mathcal{A}_i} x_{i\alpha_i} v_{i\alpha_i}(x) \tag{3}$$

where, in standard notation, $\langle v, x \rangle = v^\top x$ denotes the ordinary pairing between $v$ and $x$.

In terms of solutions, the most widely used concept in game theory is that of a *Nash equilibrium* (NE), i.e., a state $x^* \in \mathcal{X}$ such that

$$u_i(x^*) \geq u_i(x_i; x_{-i}^*) \quad \text{for all } x_i \in \mathcal{X}_i \text{ and all } i \in \mathcal{N}. \tag{NE}$$

Writing $\mathrm{supp}(x_i^*) = \{\alpha_i \in \mathcal{A}_i : x_{i\alpha_i}^* > 0\}$ for the support of $x_i^*$, Nash equilibria can be equivalently characterized via the variational inequality

$$v_{i\alpha_i^*}(x^*) \geq v_{i\alpha_i}(x^*) \quad \text{for all } \alpha_i^* \in \mathrm{supp}(x_i^*) \text{ and all } \alpha_i \in \mathcal{A}_i, i \in \mathcal{N}. \tag{4}$$

In turn, this characterization leads to the following taxonomy:

1. $x^*$ is called *pure* if $\mathrm{supp}(x^*) = \prod_i \mathrm{supp}(x_i^*)$ is a singleton.
2. If $x^*$ is not pure, we say that it is *mixed*; and if $\mathrm{supp}(x^*) = \mathcal{A}$, we say that it is *fully mixed*.

By definition, pure Nash equilibria are themselves pure strategies and correspond to vertices of $\mathcal{X}$; at the other end of the spectrum, fully mixed equilibria belong to the relative interior $\mathrm{ri}(\mathcal{X})$ of $\mathcal{X}$, so they are often referred to as *interior* equilibria.

Another key distinction between Nash equilibria concerns the defining inequality (NE): if this inequality is strict for all $x_i \neq x_i^*$, $i \in \mathcal{N}$, $x^*$ is called itself *strict*. Strict Nash equilibria are pure a fortiori, and they play a key role in game theory because any unilateral deviation incurs a strict loss to the deviating player; put differently, if $x^*$ is strict, *every player has a unique best response*. Taking this idea further, $x^*$ is called *quasi-strict* if (4) is strict for all $\alpha_i \in \mathcal{A}_i \setminus \mathrm{supp}(x_i^*)$, i.e., if all best responses of player $i$ are contained in $\mathrm{supp}(x_i^*)$. By a deep result of Ritzberger [51], all Nash equilibria are quasi-strict in almost all games;[4] in view of this, we will tacitly assume in the sequel that all equilibria considered are quasi-strict, a property known as "genericity" [14, 22, 31].

*Remark.* We should stress here that quasi-strict equilibria *need not be pure*: they could be partially or even fully mixed, e.g., as in the case of Stag Hunt, Rock-Paper-Scissors, Matching Pennies, the Battle of the Sexes, etc. We provide a series of illustrative examples in the supplement.

**Regret.** A key requirement in online learning is the minimization of the players' *regret*, i.e., the cumulative payoff difference between a player's mixed strategy at a given time and the player's best possible strategy in hindsight. In more detail, assuming that play evolves in continuous time $t \geq 0$, the (external) regret of a player $i \in \mathcal{N}$ relative to a sequence of play $x(t) \in \mathcal{X}$ is defined as

$$\text{Reg}_i(T) = \max_{p_i \in \mathcal{X}_i} \int_0^T [u_i(p_i; x_{-i}(t)) - u_i(x(t))] \, dt, \tag{5}$$

and we say that player $i$ has *no regret* under $x(t)$ if $\text{Reg}_i(T) = o(T)$.

**No-regret learning via regularization.** The most widely used method to achieve no-regret is the class of policies known as *follow the regularized leader* (FTRL) [56, 57]. Heuristically, at each $t \geq 0$, FTRL prescribes a mixed strategy that maximizes the players' cumulative payoff up to time $t$ minus a regularization penalty which incentivizes exploration. Formally, this is represented by the dynamics

$$y_{i\alpha_i}(t) = y_{i\alpha_i}(0) + \int_0^t v_{i\alpha_i}(x(s)) \, ds \qquad \{\text{aggregate payoffs}\}$$

$$x_{i\alpha_i}(t) = Q_{i\alpha_i}(y_i(t)) \qquad \{\text{choice of strategy}\}$$

or, in more compact notation:

$$\dot{y}(t) = v(Q(y(t))). \tag{FTRL}$$

In the above, each $y_{i\alpha_i}$ plays the role of an auxiliary "score variable" which measures the aggregate performance of the pure strategy $\alpha_i \in \mathcal{A}_i$ over time. These scores are subsequently tranformed to mixed strategies by means of a player-specific *choice map* $y_i \mapsto x_i = Q_i(y_i)$ which is defined as

$$Q_i(y_i) = \arg\max_{x_i \in \mathcal{X}_i} \{\langle y_i, x_i \rangle - h_i(x_i)\} \quad \text{for all } y_i \in \mathcal{Y}_i := \mathbb{R}^{n_i}. \tag{6}$$

In other words, $Q_i : \mathcal{Y}_i \to \mathcal{X}_i$ essentially acts as a "soft" version of the best-response correspondence $y_i \mapsto \arg\max_{x_i \in \mathcal{X}_i} \langle y_i, x_i \rangle$, suitably regularized by a convex penalty term $h_i(x_i)$. The precise assumptions regarding the *regularizer function* $h_i : \mathcal{X}_i \to \mathbb{R}$ will be discussed in detail later; for now, we provide two prototypical examples of (FTRL) that will play a major role in the sequel:

**Example 2.1** (Entropic regularization and exponential weights). One of the most widely used regularizers in online learning is the (negative) Gibbs-Shannon entropy $h_i(x_i) = \sum_{\alpha_i} x_{i\alpha_i} \log x_{i\alpha_i}$. A standard calculation then yields the so-called *logit choice map*, written in vectorized form as $\Lambda_i(y_i) = \exp(y_i)/\sum_{\alpha_i \in \mathcal{A}_i} \exp(y_{i\alpha_i})$. In turn, this leads to the *exponential weights* dynamics:

$$\begin{aligned} \dot{y}_i(t) &= v_i(x(t)), \\ x_i(t) &= \Lambda_i(y_i(t)). \end{aligned} \tag{EW}$$

The system (EW) describes the mean dynamics of the so-called *multiplicative weights update* (MWU) algorithm (or "Hedge"); for an (incomplete) account of its long history, see [2, 3, 11, 21, 29, 32, 33, 62] and references therein.

**Example 2.2** ($L^2$ regularization). Another popular choice of regularizer is the quadratic penalty $h_i(x_i) = (1/2)\|x_i\|^2$. In this case, the associated choice map is the Euclidean projector on the simplex, $\Pi_i(y_i) = \arg\min_{x_i \in \mathcal{X}_i} \|y_i - x_i\|$, which gives rise to the *Euclidean regularization dynamics*

$$\begin{aligned} \dot{y}_i(t) &= v_i(x(t)), \\ x_i(t) &= \Pi_i(y_i(t)). \end{aligned} \tag{ERD}$$

Beyond the two prototypical examples discussed above, the origin of the dynamics (FTRL) can be traced to Shalev-Shwartz & Singer [57], Nesterov [42], and, via their link to online mirror descent (OMD), all the way back to Nemirovski & Yudin [41]. Describing the history and literature surrounding these dynamics would take us too far afield, so we do not attempt it.

## 3 The fundamental dichotomy of FTRL dynamics

To connect the long-run behavior of (FTRL) to the Nash equilibria of the underlying game, we must first understand how the players' mixed strategies evolve under (FTRL). Our goal in this section is to provide some background to this question as a precursor to our analysis in Section 4. To lighten notation, we will drop in what follows the player index $i$, writing for example $x_\alpha$ instead of the more cumbersome $x_{i\alpha_i}$; we will only reinstate the index $i$ if absolutely necessary to avoid confusion.

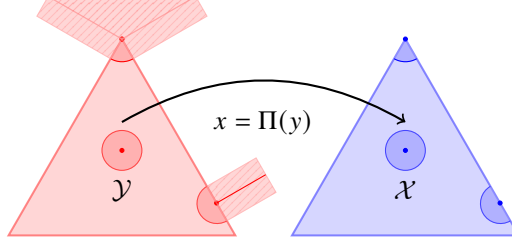

**Figure 1:** The inverse images of neighborhoods of different points in $\mathcal{X}$ under the Euclidean choice map $Q = \Pi$.

**3.1. Scores vs. strategies.** To begin, we note that (FTRL) exhibits a unique duality: on the one hand, the variables of interest are the players' mixed strategies $x(t) \in \mathcal{X}$; on the other, the dynamics (FTRL) evolve in the space $\mathcal{Y}$ of the players' score variables $y(t)$. Mixed strategies are determined by the corresponding scores via the players' choice maps $y \mapsto x = Q(y)$, but this is not a two-way street: as we explain below, the map $Q\colon \mathcal{Y} \to \mathcal{X}$ is *not invertible*, so obtaining an autonomous dynamical system on the strategy space $\mathcal{X}$ is a delicate affair. In the general case, invoking standard arguments from convex analysis [7, 52] we have $y(t) \in \nabla h(x(t)) + \mathrm{PC}(x(t))$, where

$$\mathrm{PC}(x) = \{y \in \mathcal{Y} : y_\alpha \geq y_\beta \text{ for all } \alpha \in \mathrm{supp}(x), \beta \in \mathcal{A}\} \tag{7}$$

denotes the polar cone to $\mathcal{X}$ at $x$.[5]

In the entropic case of Example 2.1, the logit choice map $Q = \Lambda$ only returns *fully mixed* strategies since $\exp(y) > 0$. In the relative interior $\mathrm{ri}(\mathcal{X})$ of $\mathcal{X}$, we have by Eq. (7) that $\mathrm{PC}(x) = \{(t, \ldots, t) : t \in \mathbb{R}\}$. As a result, $\Lambda$ is not surjective; however, up to a multiple of $(1, \ldots, 1)$, it is *injective*. On the other hand, in the Euclidean framework of Example 2.2, the choice map $Q = \Pi$ can also return non-fully mixed strategies. Both Eq. (7) and Fig. 1 show that on the boundary $\mathrm{PC}(x)$ is strictly larger compared to the interior. Thus $\Pi$ is surjective but not injective, even modulo a subspace of $\mathcal{Y}$.

The key obstacle to mapping the dynamics (FTRL) to $\mathcal{X}$ is the lack of injectivity of $Q$. In turn, this allows us to make two key observations: *(i)* there is an important split in behavior between boundary and interior states; and *(ii)* this split is linked to whether the underlying choice map is surjective or not. We elaborate on this below.

**3.2. The steep/non-steep dichotomy.** The lack of injectivity of $\Lambda$ on $\mathrm{ri}(\mathcal{X})$ is a technical artifact of the sum-to-one constraints of the strategy probabilities: knowing all but one of the strategy probabilities we can easily recover the remaining one. Thus the $\mathcal{Y}$ space, having the same number of coordinates as the $\mathcal{X}$ space, also contains redundant information. With an appropriate projection we can remove this redundancy and restore injectivity in the interior, deriving the dynamics of $x(t)$ on $\mathcal{X}$. Making this argument precise for the entropic case of Example 2.1, we obtain the *replicator dynamics*:

$$\dot{x}_\alpha = x_\alpha [v_\alpha(x) - u(x)]. \tag{RD}$$

On the other hand, this is not enough for the Euclidean framework of Example 2.2. When trajectories approach $\mathrm{bd}(\mathcal{X})$, the positivity constraints $x_i \geq 0$ kick in finite time. Unlike the sum-to-one constraints of the previous case, these cannot be resolved with a dimensionality reduction so we cannot obtain a well-posed dynamical system on $\mathcal{X}$ as above. This problem can only be temporarily avoided for time intervals where $\mathrm{supp}(x(t))$ remains constant. For these intervals $x(t)$ can be shown to satisfy the *projection dynamics* [34]

$$\dot{x}_\alpha = v_\alpha(x) - |\mathrm{supp}(x)|^{-1} \sum_{\beta \in \mathrm{supp}(x)} v_\beta(x) \quad \text{if } \alpha \in \mathrm{supp}(x). \tag{PD}$$

In contrast to the replicator dynamics, different trajectories of (PD) can merge or split any number of times, and they may transit from one face of $\mathcal{X}$ to another in finite time [34, 35].

The two cases above are not just conveniently chosen examples, but archetypes of the fundamentally different behaviors that can be observed under (FTRL) for different regularizers. As we discuss in the

supplement, this polar split is intimately tied to the behavior of the derivatives of $h$ at the boundary of $\mathcal{X}$. To formalize this, we say that $h$ is *steep* if $\|\nabla h(x)\| \to \infty$ whenever $x \to \mathrm{bd}(\mathcal{X})$; by contrast, if $\sup_{x \in \mathcal{X}} \|\nabla h(x)\| < \infty$, we say that $h$ is *non-steep*. Thus, in terms of our examples, the negentropy function of Example 2.1 is the archetype for steep regularizers, while the $L^2$ penalty of Example 2.2 is the non-steep one. The split between steep and non-steep dynamics may then be stated as follows:

1. If $h$ is steep, the mixed-strategy trajectories $x(t) = Q(y(t))$ carry all the information required to predict the evolution of the system; in particular, $x(0)$ fully determines $x(t)$ for all $t \geq 0$, and $x(0)$ remains fully mixed for all time.

2. If $h$ is non-steep, the trajectories $x(t) = Q(y(t))$ do not fully capture the state of the system: $x(0)$ *does not* determine $x(t)$ for all $t \geq 0$, and even the times when $x(t)$ changes support cannot be anticipated by knowing $x(0)$ alone. For concision, we defer the precise statement and proof of this dichotomy to the paper's supplement.

## 4 Convergence analysis and results

We now turn to the equilibrium convergence properties of (FTRL). The central question that we seek to address here is the following: *Which Nash equilibria can be stable and attracting under* (FTRL)? *Are all equilibria created equal in that regard?*

**4.1. Notions of stability.** At a high level, a point is (*a*) *stable* when every trajectory that starts nearby remains nearby; and (*b*) *attracting* when it attracts all trajectories that start close enough. Already, this heuristic shows that defining these notions for (FTRL) is not straightforward: the target points are strategy profiles in $\mathcal{X}$, while the dynamics (FTRL) evolve in the dual space $\mathcal{Y}$. When $h$ is *steep*, we can define an equivalent presentation of (FTRL) on $\mathcal{X}$, so this problem can be circumvented by working solely with mixed strategies; however, when $h$ is *non-steep*, this is no longer possible and we need to navigate carefully between $\mathcal{X}$ and $\mathcal{Y}$. In view of this, we have the following definitions:

- $x^* \in \mathcal{X}$ is *stable* if, for every neighborhood $U$ of $x^*$ in $\mathcal{X}$, there exists a neighborhood $U'$ of $x^*$ such that $x(t) = Q(y(t)) \in U$ for all $t \geq 0$ whenever $x(0) = Q(y(0)) \in U'$.

- $x^* \in \mathcal{X}$ is *attracting* if there exists a neighborhood $U$ of $x^*$ in $\mathcal{X}$ such that $x(t) = Q(y(t)) \to x^*$ whenever $x(0) = Q(y(0)) \in U$.

- $x^* \in \mathcal{X}$ is *asymptotically stable* if it is both stable and attracting.

For obvious reasons, asymptotic stability is the "gold standard" for questions pertaining to equilibrium convergence and it will be our litmus test for the appropriateness of an equilibrium $x^* \in \mathcal{X}$ as an outcome of play. Specifically, if a Nash equilibrium is not asymptotically stable under (FTRL), it is not reasonable to expect a no-regret learner to converge to it, meaning in turn that it cannot be justified as an end-state of the players' learning process. We expound on this below.

**4.2. Volume preservation.** A key observation regarding asymptotic stability is that neighborhoods of initial conditions near an asymptotically stable point should "contract" over time, eventually shrinking down to the point in question. Our first result below provides an apparent contradiction to this principle: it shows that volume is preserved under (FTRL), *irrespective of the underlying game*.

**Proposition 4.1.** *Let $\mathcal{R}_0 \subseteq \mathcal{Y}$ be a set of initial conditions for* (FTRL) *and let $\mathcal{R}_t = \{y(t) : y(0) \in \mathcal{R}_0\}$ denote its evolution under* (FTRL) *after time $t \geq 0$. Then,* $\mathrm{vol}(\mathcal{R}_t) = \mathrm{vol}(\mathcal{R}_0)$.

Proposition 4.1 (which we prove in the supplement through an application of Liouville's formula) is surprising in its universality as it holds for *all games* and *all instances* of (FTRL). As such, it provides a blanket generalization of the well-known volume-preserving property for the replicator dynamics established by Akin [1], as well as subsequent results for zero-sum games [38].

**4.3. Instability of fully mixed equilibria.** As stated above, the volume-preserving property of (FTRL) would seem to suggest that no strategy can be asymptotically stable. However, this is a figment of the duality between strategy and score variables: a mixed strategy orbit $x(t) = Q(y(t))$ could converge in $\mathcal{X}$, even though the corresponding dual orbit $y(t)$ diverges in $\mathcal{Y}$ (for an illustration, see Fig. 2 above). This again brings into sharp contrast the behavior of (FTRL) at the boundary of $\mathcal{X}$ versus its behavior at the interior. Our first instability result below shows that the volume-preserving property of (FTRL) rules out the stability of *any* fully mixed equilibrium, in *any* game:

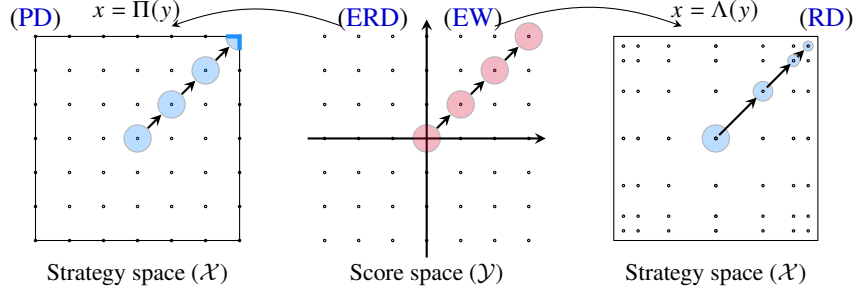

**Figure 2:** The duality between scores and strategies under (FTRL): the dynamics are volume-preserving in $\mathcal{Y}$, but a volume of initial conditions could either collapse in finite time (in the Euclidean case, left), or shrink asymptotically (in the logit case, right). This is due to the vastly different geometric properties of each system.

**Theorem 4.2.** *A fully mixed Nash equilibrium cannot be asymptotically stable under* (FTRL).

The main idea of the proof of Theorem 4.2 relies on a tandem application of Proposition 4.1 together with the dimensionality reduction idea we discussed for the entropic case in Section 3. In the resulting quotient space, the inverse image of an interior point $x^* \in \mathrm{ri}(\mathcal{X})$ is a single point and the induced dynamics remain volume-preserving. If $x^*$ is asymptotically stable, a limit point argument rules out the possibility of a trajectory entering and exiting a small neighborhood of its preimage infinitely many times. At the same time, Lyapunov stability and volume preservation imply that the dynamics are locally recurrent. This contradicts the transient property established above and proves that $x^*$ cannot be asymptotically stable; the details involved in making these arguments precise are fairly intricate, so we defer the proof of Theorem 4.2 to the supplement.

This universal instability result has significant implications as it provides a dynamic justification of the fragility of fully mixed Nash equilibria. Theorem 4.2 illustrates this principle through the lens of regret minimization: any deviation from a fully mixed equilibrium invariably creates an opportunity that can be exploited by a no-regret learner. When every player adheres to such a policy, this creates a vicious cycle which destroys any chance of stability for fully mixed equlibria.

**4.4. The case of partially mixed equilibria.**    Taking this premise to its logical extreme, a natural question that arises is whether this instability persists as long as even a *single* player employs a mixed strategy at equilibrium. In the previous case, after the dimensionality reduction argument we described in Section 3, neighborhoods of fully mixed equilibria in the space of strategies ($\mathcal{X}$) correspond to sets of finite volume in the space of payoffs ($\mathcal{Y}$). On the contrary, the case of *partially* mixed equilibria is much more complex because neighborhoods of points on the boundary of $\mathcal{X}$ correspond to sets of *infinite* volume in the space of payoffs – and this, even after dimensionality reduction (cf. Fig. 1). Because of this, volume preservation arguments cannot rule out asymptotic stability of Nash equilibria lying at the boundary of the strategy space: indeed, pure Nash equilibria also lie on the boundary but they *can* be asymptotically stable [14, 17, 34, 35].

In view of the above, it is not a priori clear whether partially mixed equilibria would behave more like pure or fully mixed ones – or if no conclusion can be drawn whatsoever. Our next result shows that the dynamics of FTRL represent a very sharp selection mechanism in this regard:

**Theorem 4.3.** *Only strict Nash equilibria can be asymptotically stable under* (FTRL).

**Corollary 4.4.** *If $x^*$ is partially mixed, it cannot be asymptotically stable under* (FTRL).

Viewed in isolation, Theorem 4.2 would seem to be subsumed by Theorem 4.3, but this is not so: the former plays an integral role in the proof of the latter, so it cannot be viewed as a special case. In more detail, the proof of Theorem 4.3 builds on Theorem 4.2 along two separate axes, depending on whether the underlying regularizer is steep or not:

1. In the steep case, as we discussed in Section 3 there is a well-posed dynamical system on $\mathcal{X}$. As we show in the supplement, each face of $\mathcal{X}$ is forward-invariant in this system, so $x^*$ must also be asymptotically stable when constrained to the face $\mathcal{X}^*$ of $\mathcal{X}$ spanned by $\mathrm{supp}(x^*)$. The conclusion of Theorem 4.3 then follows by noting that $x^*$ is interior in $\mathcal{X}^*$ and applying Theorem 4.2 to the restriction of the underlying game to $\mathcal{X}^*$.

2. The non-steep case is considerably more difficult because (FTRL) no longer induces a well-posed system on $\mathcal{X}$. In lieu of this, by examining the finer structure of the inverse image of $x^*$, it is possible to show the following: for every small enough compact neighborhood $\mathcal{K}$ of $x^*$ in $\mathcal{X}$, there exists a finite time $\tau_{\mathcal{K}} \geq 0$ such that $\text{supp}(x(t)) = \text{supp}(x^*)$ for all $t \geq \tau_{\mathcal{K}}$ whenever $x(0) \in \mathcal{K}$. As it turns out, the dynamics after $t \geq \tau_{\mathcal{K}}$ locally coincide with the mixed strategy dynamics of (FTRL) applied to the restriction of the underlying game to the face $\mathcal{X}^*$ of $\mathcal{X}$ spanned by $x^*$. Since $x^*$ is a fully mixed equilibrium in this restricted game, it cannot be asymptotically stable.

**4.5. Stable limit sets.** We conclude our analysis with a result concerning more general behaviors whereby the dynamics of FTRL do not converge to a point, but to a more general *invariant set* – such as a chain of stationary points interconnected by solution orbits, a structure known as a *heteroclinic cycle* [see e.g., 26, 55, and references therein]. As an example, in the case of two-player zero-sum games with a fully mixed equilibrium, it is known that the trajectories of (FTRL) form periodic orbits (cycles). However, these orbits are *not* asymptotically stable: if the initialization of the FTRL dynamics is slightly perturbed, the resulting trajectory will be a different periodic orbit, which does not converge to the first (in the language of dynamical systems, the cycles observed in zero-sum games are not *limit cycles*). We are thus led to the following natural question:

*What type of invariant structures can arise as stable limits of* (FTRL)*?*

To state this question formally, we will require the setwise version of asymptotic stability: a set $\mathcal{S}$ is called *asymptotically stable* under (FTRL) if *a)* all orbits $x(t) = Q(y(t))$ of (FTRL) that start sufficiently close to $\mathcal{S}$ remain close; and *b)* all orbits that start nearby eventually converge to $\mathcal{S}$. Then, focusing on the case of steep dynamics to avoid more complicated statements, we have:

**Theorem 4.5.** *Every asymptotically stable set of steep* (FTRL) *contains a pure strategy.*

The proof of Theorem 4.5 relies on an "infinite descent" argument whereby the faces of $\mathcal{X}$ that intersect with $\mathcal{S}$ are eliminated one-by-one, until only pure strategies remain as candidate elements of $\mathcal{S}$ with minimal support; we provide the details in the supplement.

The importance of Theorem 4.5 lies in that it provides a succinct criterion for identifying possible attracting sets of (FTRL). Indeed, by Conley's decomposition theorem (also known as the "fundamental theorem of dynamical systems") [16], the flow of (FTRL) in an arbitrary game decomposes into a chain recurrent part and an attracting part (see [45, 46] for several examples/discussion in the case of replicator dynamics). The recurrent part is exemplified by the periodic orbits that arise in zero-sum games with an interior equilibrium (there are no attractors in this case) [38]. Theorem 4.5 goes a long way to showing that the attracting part of (FTRL) always intersects the *extremes* of the game's strategy space – i.e., the players' set of *pure* strategies. A special case of Theorem 4.5, in the case of replicator dynamics, was employed in [44] as a step in the definition of new, dynamics/decomposition-based solution concepts. Formalizing the exact form of this decomposition in arbitrary games is an open direction for future research with far-reaching implications for the theory of online learning in games.

## 5   Concluding remarks

The well known universal existence theorem for (mixed) Nash equilibria in general games has been very influential not only from a mathematics perspective but also from a public policy one as it seems to suggest that there is no inherent tension in any societal setting between the single-minded pursuit of individual profits and societal stability. Nash equilibria satisfy both desiderata simultaneously. Thus, there is in principle no need for centralized intervention and guidance as market forces will converge upon such a solution.

Our results present an argument in the opposite direction. Unless the game has a pure Nash equilibrium, which is definitely not satisfied in numerous strategic interactions, then societal systems do not self-stabilize, even if they are driven by our most effective payoff seeking dynamics, i.e., gradient learning and its follow-the-regularizer-leader variants. Exploring the tradeoffs between individual optimality and societal stability is thus a much more subtle issue than it first meets the eye, and we hope that we inspire follow-up work that can elucidate these questions further.

## Acknowledgments

This research was partially supported by the COST Action CA16228 "European Network for Game Theory" (GAMENET), the French National Research Agency (ANR) under grant ALIAS, and the Onassis Foundation under Scholarship ID: F ZN 010-1/2017-2018.

E.V. Vlatakis-Gkaragkounis is grateful to be supported by NSF grants CCF-1703925, CCF-1763970, CCF-1814873, CCF-1563155, and by the Simons Collaboration on Algorithms and Geometry.

T. Lianeas is supported by the Hellenic Foundation for Research and Innovation (H.F.R.I.) under the "First Call for H.F.R.I. Research Projects to support Faculty members and Researchers and the procurement of high-cost research equipment grant", project BALSAM, HFRI-FM17-1424.

P. Mertikopoulos is grateful for financial support by the French National Research Agency (ANR) in the framework of the "Investissements d'avenir" program (ANR-15-IDEX-02), the LabEx PERSY-VAL (ANR-11-LABX-0025-01), and MIAI@Grenoble Alpes (ANR-19-P3IA-0003).

G. Piliouras gratefully acknowledges AcRF Tier-2 grant (Ministry of Education – Singapore) 2016-T2-1-170, grant PIE-SGP-AI-2018-01, NRF2019-NRF-ANR095 ALIAS grant and NRF 2018 Fellowship NRF-NRFF2018-07 (National Research Foundation Singapore).

## Broader Impact

This is a theoretical work which does not present any foreseeable societal consequence.

## Footnotes

[2]Interestingly, Akin's result was established under a special *non-Euclidean* volume form on the game's *strategy* space, a fact which made any attempts at generalization particularly elusive.

[3]We are using here the standard game-theoretic shorthand $(x_i; x_{-i}) := (x_1, \ldots, x_i, \ldots, x_N)$ to highlight the strategic choice of a given player $i \in \mathcal{N}$ versus that of the player's opponents $\mathcal{N}_{-i} := \mathcal{N} \setminus \{i\}$.

[4]Specifically, on a set which is open and dense (and hence of full measure) in the space of all games.

[5]In particular, for all $y \in \mathrm{PC}(x)$, we have $y_\alpha = y_\beta$ whenever $\alpha, \beta \in \mathrm{supp}(x)$. The similarity of this condition to the characterization (4) of Nash equilibria is not a coincidence: $x^*$ is a Nash equilibrium of $\Gamma$ if and only if $v(x^*) \in \mathrm{PC}(x^*)$ [14, 35].

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
