[Supplementary Material]

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

[6]Here, "almost" means that the set of such states has full Lebesgue measure.

[7]A smooth map $\Phi \colon \mathcal{X} \times [0, \infty) \to \mathcal{X}$ is called a *semiflow* if $\Phi_0(x) = x$ and $\Phi_{t+s}(x) = \Phi_s(\Phi_t(x))$ for all $x \in \mathcal{X}$ and all $t, s \geq 0$. Heuristically, $\Phi_t(x) \equiv \Phi_t(x)$ describes the trajectory of the dynamical system starting at $x$.

[8]Specifically, $\Pi_i(y_i) = \Pi_i(y'_i)$ if and only if $y'_{i\alpha_i} = y_{i\alpha_i} + c$ for some $c \in \mathbb{R}$ and all $\alpha_i \in \mathcal{A}_i$.

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

## A  An ontology of Nash equilibria: representative examples

In the archetypal game of Prisoner's Dilemma (left), it is easy to check that the unique Nash equilibrium is the mutual betrayal which is strict (and hence pure). On the other hand, Matching Pennies (right) is an example of a zero-sum game whose unique Nash equilibrium is fully mixed but still quasi-strict (since all strategies present in its support are unilateral best responses to it). We mention the above to clarify that quasi-strict *does not mean* pure equilibria and includes also the fully mixed Nash equilibrium; the terminology is, perhaps, unfortunate, but otherwise deeply entrenched in the game-theoretic literature [22].

<table>
<tr><td></td><td></td><td colspan="2" align="center">Player $Y$</td><td></td><td></td><td colspan="2" align="center">Player $Y$</td></tr>
<tr><td></td><td></td><td align="center">$B$</td><td align="center">$S$</td><td></td><td></td><td align="center">$H$</td><td align="center">$T$</td></tr>
<tr><td rowspan="2">Player $X$</td><td>$B$</td><td>$(3,3)$</td><td>$(0,5)$</td><td rowspan="2">Player $X$</td><td>$H$</td><td>$(1,-1)$</td><td>$(-1,1)$</td></tr>
<tr><td>$S$</td><td>$(5,0)$</td><td>$(1,1)$</td><td>$T$</td><td>$(-1,1)$</td><td>$(1,-1)$</td></tr>
</table>

**Table 1:** Prisoner's Dilemma (left) & Matching Pennies (right).

## B  Basic properties of the FTRL dynamics

**B.1. Definitions from dynamical systems.**   In this appendix, we provide some general preliminaries from general topology and the theory of dynamical systems that we will use freely in the sequel.

A key notion in our analysis is that of (*Poincaré*) *recurrence*. Intuitively, a dynamical system is recurrent if, after a sufficiently long (but *finite*) time, almost every state returns arbitrarily close to the system's initial state.[6] More formally, given a dynamical system on $\mathcal{X}$ that is defined by means of a *semiflow* $\Phi \colon \mathcal{X} \times [0, \infty) \to \mathcal{X}$, we have:[7]

**Definition B.1.** A point $x \in \mathcal{X}$ is said to be *recurrent* under $\Phi$ if, for every neighborhood $U$ of $x$ in $\mathcal{X}$, there exists an increasing sequence of times $t_n \uparrow \infty$ such that $\Phi_{t_n}(x) \in U$ for all $n$. Moreover, the flow $\Phi$ is called (*Poincaré*) *recurrent* if, for every measurable subset $A$ of $\mathcal{X}$, the set of recurrent points in $A$ has full measure.

The above definition directly implies that the flow $\Phi_t(x)$ from a recurrent point $x$ cannot converge to any $x' \neq x$. Poincaré's recurrence theorem gives sufficient condition for the existence of such points.

**Theorem B.1** (Poincaré Recurrence Theorem). *If a flow $\Phi$ preserves volume and its orbits are bounded, then almost every point is recurrent under $\Phi$.*

The key notion in the above formulation of the theorem is that of volume preservation: formally, a flow $\Phi$ is *volume-preserving* if $\mathrm{vol}(\Phi_t(\mathcal{R})) = \mathrm{vol}(\mathcal{R})$ for any set of initial conditions $\mathcal{R} \subseteq \mathcal{X}$. A useful condition to establish this property is via *Liouville's formula*, as stated below:

**Theorem B.2** (Liouville's formula). *Let $\Phi$ be the flow of a dynamical system with infinitesimal generator $V$, i.e., $\Phi_t(x)$ is the solution trajectory of the ordinary differential equation*

$$\frac{d}{dt} x(t) = V(x(t)) \tag{B.1}$$

*with initial condition $x(0) = x$. Then, letting $\mathcal{R}_t = \Phi_t(\mathcal{R})$ for an arbitrary measurable set $\mathcal{R}$, we have*

$$\frac{d}{dt} \mathrm{vol}[\mathcal{R}_t] = \int_{\mathcal{R}_t} \mathrm{div}[V(x)]\, dx \tag{B.2}$$

**Corollary B.3.** *If $V$ is incompressible over $\mathbb{R}^n$ (i.e., $\mathrm{div}\, V(x) = 0$ for all $x \in \mathbb{R}^n$), the induced flow $\Phi$ is volume-preserving.*

**B.2. Structural properties of the FTRL dynamics: the steep/non-steep dichotomy.** To proceed with our analysis, we will need to clarify the precise technical requirements for the dynamics' regularizer function $h$. These are as follows: *(i)* $h \in C_0(\mathbb{R}^n_+) \cap C_2(\mathbb{R}^n_{++})$, i.e., $h$ is continuous on $\mathbb{R}^n_+$ and two times continuously differentiable on $\mathbb{R}^n_{++}$; *(ii)* $h$ is strongly convex on $\mathcal{X}$; and *(iii)* the inverse Hessian $H(x) = \text{Hess}(h(x))^{-1}$ of $h$ admits a Lipschitz extension to all of $\mathcal{X}$ such that $z^\top H(x)z > 0$ whenever $\text{supp}(z) \supseteq \text{supp}(x)$. These conditions are purely technical in nature and they are satisfied by all the regularizers used in practice, cf. [34, 56**? ? ? ?**] and references therein. As we discussed in the main body of our paper, there is an important distinction to be made depending on whether $h$ is *steep* or *non-steep*. Formally, we say that $h$ is *steep* if $\|\nabla h(x)\| \to \infty$ whenever $x \to \text{bd}(\mathcal{X})$ and $\text{rank}(H(x)) = |\text{supp}(x)|$ for all $x$; by contrast, if $\sup_{x \in \mathcal{X}} \|\nabla h(x)\| < \infty$ and $\text{rank}(H(x)) = n$, we say that $h$ is *non-steep*. The qualititative difference in behavior between these cases is illustrated in the figure below (which shows the very different behavior of the derivates of $h$ near the boundary of the state space).

$$
\begin{bmatrix}
\text{steep} & h_1(x) & = & x \log(x) & + & (1-x)\log(1-x) \\
\text{non-steep} & h_2(x) & = & \frac{1}{2}x^2 & + & \frac{1}{2}(1-x)^2
\end{bmatrix}
$$

> The following lemma illustrates the relation between mixed strategies and score vectors and the mirror map (6) that defines the dynamics (FTRL). We focus on the perspective of an arbitrary player, say $i$, and for ease of notation we write $Q$, $x$ and $y$ instead of $Q_i$, $x_i$ and $y_i$ respectively. The lemma begins to illustrate the gulf between the steep and non-steep cases.

**Lemma B.4.** $x = Q(y)$ *if and only if there exist* $\mu \in \mathbb{R}$ *and* $\nu_\alpha \in \mathbb{R}_+$ *such that, for all* $\alpha \in \mathcal{A}$, *we have: a)* $y_\alpha = \frac{\partial h}{\partial x_\alpha} + \mu - \nu_\alpha$; *and b)* $x_\alpha \nu_\alpha = 0$ *In particular, if $h$ is steep, we have* $\nu_\alpha = 0$ *for all* $\alpha \in \mathcal{A}$.

*Proof.* Recall that

$$
Q(y) = \arg\max_{x \in \mathcal{X}} \{\langle y, x \rangle - h(x)\}
$$

$$
= \arg\max\left\{\sum_{\alpha \in \mathcal{A}} y_\alpha x_\alpha - h(x) : \sum_{\alpha \in \mathcal{A}} x_\alpha = 1 \text{ and } \forall \alpha \in \mathcal{A} : x_\alpha \geq 0\right\}
$$

The result follows by applying the Karush–Kuhn–Tucker (KKT) conditions to this optimization problem and noting that, since the constraints are affine, the KKT conditions are sufficient for optimality. Our Langragian is

$$
\mathcal{L}(x, \mu, \nu) = \left(\sum_{\alpha \in \mathcal{A}} y_\alpha x_\alpha - h(x)\right) - \mu\left(\sum_{\alpha \in \mathcal{A}} x_\alpha - 1\right) + \sum_{\alpha \in \mathcal{A}} \nu_\alpha x_\alpha
$$

where the set of constraints (i) of the statement of the lemma are the stationarity constraints, which in our case are $\nabla \mathcal{L}(x, \mu, \nu) = 0 \Leftrightarrow \nabla(\sum_{\alpha \in \mathcal{A}} y_\alpha x_\alpha - h(x)) = \mu \nabla(\sum_{\alpha \in \mathcal{A}} x_\alpha - 1) - \sum_{\alpha \in \mathcal{A}} \nu_\alpha \nabla x_\alpha$ , while

the set of constraints (ii) of the statement of the lemmas are the complementary slackness constraints. Note that complementary slackness implies that whenever $v_\alpha > 0$ whenever $\alpha \notin \mathrm{supp}(x)$. Finally, if $h$ is steep, we have $|\partial_\alpha h(x)| \to \infty$ as $x \to \mathrm{bd}(\mathcal{X})$, which implies that the KKT conditions admit a solution with $v_\alpha = 0$. ∎

> The following lemma shows that if the support of $x(t)$ does not change over a given interval of time, then the evolution of the players' mixed strategies under (FTRL) follows a certain differential equation that can be calculated explicitly. The lemma below also shows that the trajectory of play coincides with the trajectory that would have resulted if the game were constrained to the strategies present in the support of $x(t)$. Again, for ease of notation we focus on player $i$ and omit $i$ from all subscripts.

**Proposition B.5.** *Let $x(t) = Q(y(t))$ be a mixed strategy orbit of (FTRL), and let $\mathcal{T}$ be an interval over which $\mathrm{supp}(x(t))$ is constant. Then, for all $t \in \mathcal{T}$, $x(t)$ satisfies the mixed strategy dynamics:*

$$\dot{x}_\alpha = \sum_{\beta \in \mathrm{supp}(x)} \left[ H_{\alpha\beta}(x) - H_\alpha(x) H_\beta(x) \right] v_\alpha(x), \tag{FTRL-s}$$

*where $H_\alpha(x) = \left[ \sum_{\beta,\beta' \in \mathrm{supp}(x)} H_{\beta\beta'}(x) \right]^{-1/2} \sum_{\beta \in \mathrm{supp}(x)} H_{\alpha\beta}(x)$. In particular, we have the following dichotomy:*

1. *If $h$ is steep, the dynamics (FTRL-s) are* well-posed, *i.e., they admit unique global solutions from any initial condition $x \in \mathcal{X}$ (including the boundary). Moreover, the faces of $\mathcal{X}$ are forward-invariant under (FTRL-s): the support of $x(t)$ remains constant for all $t \geq 0$.*

2. *If $h$ is non-steep, the dynamics (FTRL-s) are not well-posed: solutions $x(t)$ to (FTRL-s) exist only up to a finite time, after which the support of $x(t)$ may change.*

*Proof.* For the first part of the lemma, we follow a line of reasoning due to [34]. Specifically, letting $g_\alpha(x) = \partial_\alpha h(x)$, Lemma B.4 yields

$$y_\alpha(t) = g_\alpha(t) + \mu(t), \ \forall t \in I, \forall \alpha \in \mathcal{A}^* \tag{B.3}$$

Since $y_\alpha$ and $g_\alpha$ are both smooth, so is $\mu(t)$. Thus, differentiating with respect to $t$ we get

$$\dot{y}_\alpha(t) = \sum_{\beta \in \mathcal{A}} \frac{\partial^2 h}{\partial x_\beta \partial x_\alpha} \dot{x}_\beta(t) + \dot{\mu}(t)$$

$$= \sum_{\beta \in \mathcal{A}^*} \frac{\partial^2 h}{\partial x_\beta \partial x_\alpha} \dot{x}_\beta(t) + \dot{\mu}(t)$$

since for all $t \in I$ and $\beta \in \mathcal{A} \setminus \mathcal{A}^*$, $x_\beta(t) = 0$, and thus $\dot{x}_\alpha(t) = 0$. Multiplying with the inverse of the Hessian, and omitting $t$ for brevity, we get

$$\dot{x}_\alpha = \sum_{\beta \in \mathcal{A}^*} H_{\alpha\beta} \dot{y}_\beta + \sum_{\beta \in \mathcal{A}^*} H_{\alpha\beta} \dot{\mu} \tag{B.4}$$

By the definition of the dynamics, $\dot{y}_\beta = v_\beta$ and since the support remains constant $\sum_{\alpha \in \mathcal{A}^*} \dot{x}_\alpha = 0$. Summing up Eq. (B.4) for $\alpha \in \mathcal{A}^*$ we get

$$\sum_{\alpha,\beta \in \mathcal{A}^*} H_{\alpha\beta} v_\beta + G\dot{\mu} = 0 \Leftrightarrow \dot{\mu} = -\frac{\sum_{\beta \in \mathcal{A}^*} H_\beta v_\beta}{G} \tag{B.5}$$

where $G = \sum_{\beta,\beta' \in \mathrm{supp}(x)} H_{\beta\beta'}(x)$. Substituting the latter and $\dot{y}_\beta = v_\beta$ to Eq. (B.4) we get the desired result.

For the second part of the lemma, the well-posedness of (FTRL-s) follows from the fact that $H(x)$ admits a Lipschitz continuous extension to all of $x$; moreover, by the rank assumption, the field $H_\alpha(x)$ is also Lipschitz continuous (since the denominator does not vanish; recall that $\mathrm{im}\, H(x) = \mathbb{R}^{\mathrm{supp}(x)}$). Finally, forward invariance follows by noting that, for every initial condition $x \in \mathcal{X}$, the quantity $\sum_{\alpha \in \mathrm{supp}(x)} x_\alpha(t)$ is a constant of motion (identically equal to 1), and that $\dot{x}_\alpha = 0$ whenever $x_\alpha = 0$. ∎

A key take-away from the above result is that, in the steep case, the dynamics (FTRL-s) can be seen as a patchwork of dynamical systems, each evolving on a specific face of $x$, and each a continuous extension of the other at the points where they come into contact. Neither of the above is true for the non-steep case.

## B.3. Volume preservation in $\mathcal{Y}$ and $\mathrm{ri}(\mathcal{X})$.

After restating it, we proceed to show Proposition 4.1 by simply applying Liouville's formula (Theorem B.2).

**Proposition 4.1.** *Let $\mathcal{R}_0 \subseteq \mathcal{Y}$ be a set of initial conditions for (FTRL) and let $\mathcal{R}_t = \{y(t) : y(0) \in \mathcal{R}_0\}$ denote its evolution under (FTRL) after time $t \geq 0$. Then, $\mathrm{vol}(\mathcal{R}_t) = \mathrm{vol}(\mathcal{R}_0)$.*

*Proof.* We have to show that the dynamics of (FTRL) (i.e., $\dot{y}(t) = v(Q(y(t)))$) are incompressible. For any player $i$ and any $\alpha \in \mathcal{A}_i$ we have

$$\frac{\partial v_{i\alpha}}{\partial y_{i\alpha}} = \sum_{\beta \in \mathcal{A}_i} \frac{\partial v_{i\alpha_i}}{\partial x_{i\beta}} \frac{\partial x_{i\beta}}{\partial y_{i\alpha}} = 0, \tag{B.6}$$

because $v_i$ does not depend on $x_i$. We thus obtain $\mathrm{div}_y\, v(y) = 0$, i.e., the dynamics (FTRL) are incompressible. The result then follows from Liouville's formula ∎

In the following lemma, we show that the flow defined by FTRL in the interior $\mathrm{ri}(\mathcal{X})$ of $\mathcal{X}$ is incompressible under a suitably defined measure. For that, using Liouville's formula, we first show that in the so-called $z$-space, i.e., a "slice" of the payoff space, the respective flow is incompressible. Using that, we can easily get a diffeomorphism from $\mathrm{ri}(\mathcal{X})$ to the $z$-space, we define the volume of a set in $\mathrm{ri}(\mathcal{X})$ to be the volume of the corresponding set in the $z$-space, and thus incompresssibility in the interior comes for free.

**Lemma B.6.** *There exists a measure $\mu_x$ for which the flow in the interior of $\mathcal{X}$ is incompressible, i.e., for any subset $U \subset \mathrm{ri}(\mathcal{X})$ of initial conditions, and any $t_0 \geq 0$ so that for any $0 \leq t \leq t_0$: $\Phi(U, t) \subset \mathrm{ri}(\mathcal{X})$, it is $\mu_x(U) = \mu_x(\Phi(U, t))$.*

*Proof.* First we go on to define the $z$-space. The intuition for defining and using the $z$-space can be based on Lemma B.4 which implies that for any $x \in \mathrm{ri}(\mathcal{X})$, any two corresponding points $y, y'$ in

the $y$-space differ by a constant, since for all $i$ and $\alpha_j \in \mathcal{A}_i$, $y_{i\alpha_j} = \frac{\partial h}{\partial x_{i\alpha_j}} + \mu_i$ and $y'_{i\alpha_j} = \frac{\partial h}{\partial x_{i\alpha_j}} + \mu'_i$ for some $\mu_i$ and $\mu'_i$ (recall $x \in \mathrm{ri}(\mathcal{X})$ implies $v_{i\alpha_j} = 0$). Thus, all $y$'s that correspond to an $x \in \mathrm{ri}(\mathcal{X})$ form an equivalent class. For each class, we pick as representative the $y$ in the class that has $0$ in some specific coordinate $\hat{\alpha}_i$, for every player $i$. The set of representatives form the $z$-space and there is a a one to one correspondence of points of $\mathrm{ri}(\mathcal{X})$ to points in the $z$-space which moreover can be used to define an incompressible flow in $\mathrm{ri}(\mathcal{X})$.

So, for a *benchmark* strategy $\hat{\alpha}_i \in \mathcal{A}_i$ for every player $i \in \mathcal{N}$ and for all $\alpha \in \mathcal{A}_i \backslash \{\hat{\alpha}_i\} \equiv \hat{\mathcal{A}}_i$ consider the corresponding score differences

$$z_{i\alpha} = y_{i\alpha} - y_{i\hat{\alpha}_i}. \tag{B.7}$$

Obviously, $z_{i\alpha} = y_{i\alpha_i} - y_{i\hat{\alpha}_i}$ is identically zero so we can ignore it in the above definition. In so doing, we obtain a linear map $\Pi_i \colon \mathbb{R}^{\mathcal{A}_i} \to \mathbb{R}^{\hat{\mathcal{A}}_i}$ sending $y_i \mapsto z_i$; aggregating over all players, we also write $\Pi$ for the product map $\Pi = (\Pi_1, \ldots, \Pi_N)$ sending $y \mapsto z$. For posterity, note that this map is surjective but *not* injective,[8] so it does not allow us to recover the score vector $y$ from the score difference vector $z$.

Now, under FTRL, the score differences (B.7) evolve as

$$\dot{z}_{i\alpha} = v_{i\alpha}(x(t)) - v_{i\hat{\alpha}_i}(x(t)). \tag{B.8}$$

Our first step below is to show that (B.8) constitutes a well-defined dynamical system on $z$ as long as the correpsonding $x$'s remain in $\mathrm{ri}(\mathcal{X})$.

To do so, consider the reduced mirror map $\hat{Q}_i \colon \mathbb{R}^{\hat{\mathcal{A}}_i} \to \mathcal{X}_i$ defined as

$$\hat{Q}_i(z_i) = Q_i(y_i) \tag{B.9}$$

for some $y_i \in \mathbb{R}^{\mathcal{A}_i}$ such that $\Pi_i(y_i) = z_i$. That such a $y_i$ exists is a consequence of $\Pi_i$ being surjective; furthemore, that $\hat{Q}_i(z_i)$ is well-defined is a consequence of the fact that $Q_i$ is invariant on the fibers of $\Pi_i$. Indeed, by construction, and as long as the corresponding $x$'s remain in $\mathrm{ri}(\mathcal{X})$ we have $\Pi_i(y_i) = \Pi_i(y'_i)$ if and only if $y'_{i\alpha} = y_{i\alpha} + c$ for some $c \in \mathbb{R}$ and all $\alpha \in \mathcal{A}_i$. Hence, by the definition of $Q_i$, we get

$$\begin{aligned} Q_i(y'_i) &= \arg\max_{x_i \in \mathcal{X}_i} \left\{ \langle y_i, x_i \rangle + c \sum_{\alpha \in \mathcal{A}_i} x_{i\alpha} - h_i(x_i) \right\} \\ &= \arg\max_{x_i \in \mathcal{X}_i} \left\{ \langle y_i, x_i \rangle - h_i(x_i) \right\} = Q_i(y_i), \end{aligned} \tag{B.10}$$

where we used the fact that $\sum_{\alpha \in \mathcal{A}_i} x_{i\alpha} = 1$. The above shows that $Q_i(y'_i) = Q_i(y_i)$ if and only if $\Pi_i(y_i) = \Pi_i(y'_i)$, so $\hat{Q}_i$ is well-defined. Letting $\hat{Q} \equiv (\hat{Q}_1, \ldots, \hat{Q}_N)$ denote the aggregation of the players' individual mirror maps $\hat{Q}_i$, it follows immediately that $Q(y) = \hat{Q}(\Pi(y)) = \hat{Q}(z)$ by construction.

Hence, the dynamics (B.8) may be written as

$$\dot{z} = V(z), \tag{B.11}$$

where

$$\nu_{i\alpha}(z) = v_{i\alpha}(\hat{Q}_i(z)) - v_{i\hat{\alpha}_i}(\hat{Q}_i(z)). \tag{B.12}$$

These dynamics obviously constitute an autonomous system.

Next, we show incompressibiity of the $z$-space. Indeed, for all $\alpha \in \mathcal{A}$ we have

$$\frac{\partial \nu_{i\alpha}}{\partial z_{i\alpha}} = \sum_{\beta \in \mathcal{A}_i} \frac{\partial \nu_{i\alpha_i}}{\partial x_{i\beta}} \frac{\partial x_{i\beta}}{\partial z_{i\alpha}} = 0, \tag{B.13}$$

because $v_i$ does not depend on $x_i$. We thus obtain $\nabla_z \cdot V(z) = 0$, i.e., the dynamics (B.11) are incompressible.

For the last step, for a set $A \subset \mathrm{ri}(\mathcal{X})$ define $\mu_x(A) := \mu_z(Q^{-1}(A))$, where $\mu_z$ is the Lebesgue measure in the $z$-space. Then for any $U \subset \mathrm{ri}(\mathcal{X})$, as long as $\Phi(U, t)$ remains in $\mathrm{ri}(\mathcal{X})$, it is

$$\mu_x(U) = \mu_z(Q^{-1}(U)) = \mu_z(Q^{-1}(\Phi(U, t))) = \mu_x(\Phi(U, t))$$

as needed. ∎

## C   Proof of Theorem 4.2

Below we show that there are no asymptotically stable sets (or points) in $\mathrm{ri}(\mathcal{X})$. Indeed, if this were the case, there would be a full-measure set of initial conditions outside the asymptotically stable set $A^*$ that converges to $A^*$, while at the same time its trajectories are bounded (by stability). This contradicts Poincaré's recurrence theorem, because the flow in $\mathrm{ri}(\mathcal{X})$ is volume-preserving by Lemma B.6.

**Theorem C.1.** *Let $A^*$ be a closed set of $\mathrm{ri}(\mathcal{X})$. Then $A^*$ is not asymptotically stable under FTRL.*

*Proof.* To reach a contradiction, let $A^*$ be an asymptotically stable set, i.e., attracting and Lyapunov stable, belonging in $\mathrm{ri}(\mathcal{X})$. Since $A^*$ is attracting, there exists a neighborhood $U$ of $A^*$ all points of which converge to $A^*$. Without loss of generality, since $A^*$ is closed, we may assume that $U$ lies in $\mathrm{ri}(\mathcal{X})$, and its closure is disjoint from the boundary of $\mathcal{X}$.

Now, since $A^*$ is Lyapunov stable, there exists some neighborhood $U_0$ of $A^*$ so that whenever $x(0) \in U_0$, $x(t) \in U$. Pick some $x_0 \in U_0 \setminus A^*$. Since $A^*$ is closed and $U_0$ is open, there is a small enough neighborhood $E$ of $x_0$ so that all points of $E$ lie inside $U_0 \setminus A^*$. By Lyapunov stability, for all $t \geq 0$, $\Phi(E, t) \subseteq U$. But then the set $E_\infty = \cup_{t \geq 0} \Phi(E, t)$ is bounded, having positive measure that does not change over time (Lemma B.6). Therefore it is a Poincaré recurrent set. But this means that all but a measure zero set of initializations in $E$ lead to recurrent trajectories that return infinitely often to $E$. Picking $E$ to be bounded away from $A^*$ (which is a closed set) we conclude that there are points in $E$ (and thus $U$) that do not converge to $A^*$, a contradiction. ∎

Now, given that any singleton set $\{x\}$, $x \in \mathcal{X}$, is closed, the above yields:

**Theorem C.2.** *There are no asymptotically stable points in* $\mathrm{ri}(\mathcal{X})$

Theorem 4.2 (restated below) then follows as a corollary.

**Theorem 4.2.** *A fully mixed Nash equilibrium cannot be asymptotically stable under* (FTRL).

# D   Proof of Theorem 4.3: the non-steep case

Our goal in this appendix is to provide the proof of Theorem 4.3, which we restate below for convenience:

**Theorem 4.3.** *Only strict Nash equilibria can be asymptotically stable under* (FTRL).

Because of the fundamental dichotomy between steep and non-steep FTRL dynamics, we will break the proof in two cases, treating here the non-steep regime; the steep case will be proved in Appendix E as a consequence of a more general result. The fundamental distinction between the two cases is that, in the non-steep regime, the mixed-strategy dynamics of (FTRL) could change support infinitely many times, which means that the type of volume-preservation arguments employed in the previous section cannot work (because the corresponding preimages in the $z$-space could have infinite volume; see below for a graphical illustration). However, as we show below, this "change of support" is a blessing in disguise: if $x^*$ is asymptotically stable, nearby trajectories will end up employing only those strategies present in $x^*$ in finite time.

**(PGD)**
**Payoff Space Representation**          $y = Q(x)$          **State Space Representation**

Collapse of Volume Preservation          $bd(\mathcal{X})$

$x = Q^{-1}(y)$

> The following lemma shows that, for generic games, if the underlying regularizer is non-steep, all trajectories starting near an asymptotically stable point $x^*$ attain the face of $x^*$ in some uniform, finite time. The intuition for this is that, generically, for any player $i$, the coordinates of $y_i$ that correspond to the support of $x_i^*$ increase with a "speed" that is uniformly higher than those strategies not supported in $x^*$. The regularizer of player $i$ could possibly act in favor of the coordinates that do not belong to the support, but in a bounded way, since it is non-steep. Thus, there is a time after which the coordinates of $y_i$ corresponding to the support are bigger enough than the other coordinates, so that the mirror map $Q_i$ keeps returning a point with support equal to the support of $x_i^*$.

**Lemma D.1.** *Let $x^*$ be an asymptotically stable equilirium of a generic finite game $\Gamma$, with the regularizers used, being non-steep. For any neighborhood $U$ of $x^*$, there exists a neighborhood $U_0$ of $x^*$ and a finite time $T_0$ such that if $x(t) = Q(y(t))$ is an orbit of FTRL starting at $x(0) \in U_0$, then $\mathrm{supp}(x(t)) = \mathrm{supp}(x^*)$ for all $t \geq T_0$.*

*Proof.* By the genericity assumption, all Nash equilibria are quasi-strict. Clearly we have that for any player $i$, $u_{i\alpha}(x^*) > u_{i\beta}(x^*)$ for all $\alpha \in \mathrm{supp}(x_i^*) \equiv \mathcal{A}_i^*$ and $\beta_i \notin \mathrm{supp}(x_i^*)$. Thus, by continuity there exists some neighborhood $U$ of $x^*$ and a $c > 0$ so that for any $x \in U$ and any player $i$, $u_{i\alpha}(x) > u_{i\beta}(x) + c$ for all $\alpha \in \mathcal{A}_i^*, \beta \in \mathcal{A}_i \setminus \mathcal{A}_i^*$. Additionally, we can choose $U$ small enough so that for all $x \in U$, $\mathrm{supp}(x^*) \subseteq \mathrm{supp}(x)$. Since $x^*$ is asymptotically stable there exists a neighborhood $U_0$ of $x^*$ so that $x(t) \in U$ for all $t$ whenever $x(0) \in U_0$, and $\lim_{t \to \infty} x(t) = x^*$.

Consider some $x(0) \in U_0$ and let $\alpha \in \mathcal{A}^*, \beta \in \mathcal{A} \setminus \mathcal{A}^*$. For ease of notation in the following we focus on the perspective of an arbitrary player, say $i$, and omit $i$ from the subscripts.

By Lemma B.4, for any $t \geq 0$ there exist a $\mu(t)$ and non negative $\nu_\alpha(t)$'s so that

$$y_\alpha(t) = g_\alpha(x(t)) + \mu(t) \qquad\qquad \forall \alpha \in \mathcal{A}^*$$
$$y_\alpha(t) = g_\alpha(x(t)) + \mu(t) - \nu_\beta(t) \qquad\qquad \forall \beta \in \mathcal{A} \setminus \mathcal{A}^*$$

since, by complementary slackness, $\nu_\alpha(t) = 0$, whenever $x_\alpha(t) > 0$. Subtracting we get

$$y_\alpha(t) - y_\beta(t) = g_\alpha(x(t)) - g_\beta(x(t)) + \nu_\beta(t) \leq \nu_\beta(t) + G \tag{D.1}$$

with the inequality following, for some constant $G$, by $h$ being non-steep.

On the other hand by the definition of the dynamics, using Eq. (D.1) and that $u_\alpha(x(t)) > u_\beta(x(t)) + c$, for all $t$ (since $x(0) \in U_0$), we get

$$y_\alpha(t) - y_\beta(t) = y_\alpha(0) - y_\beta(0)) + \int_0^t [u_\alpha(x(s)) - u_\beta(x(s))]ds$$
$$> g_\alpha(x(0)) - g_\beta(x(0)) + \nu_\beta(0) + ct$$
$$\geq ct + \nu_\beta(0) - G$$

with the last inequality following again by $h$ being non-steep. Combining the latter with Eq. (D.1), and since $\nu_\beta(0) \geq 0$, we get

$$\nu_\beta(t) + G > ct + \nu_\beta(0) - G \Rightarrow \nu_\beta(t) > ct - 2G$$

which implies that for $t \geq \frac{2G}{c}$ it is $\nu_\beta(t) > 0$. This in turn, by complementary slackness, yields $x_\beta(t) = 0$ for all $t \geq \frac{2G}{c}$, implying $\mathrm{supp}(x(t)) \subseteq \mathrm{supp}(x^*)$. By the choice of $U$ and since $\forall t : x(t) \in U$ we have $\mathrm{supp}(x(t)) \supseteq \mathrm{supp}(x^*)$ and thus setting $T_0 = \frac{2G}{c}$ proves the claim, since the above holds for any player $i$. ∎

> The main result of this section is the following theorem that covers the non-steep case, stating that for generic games at an asymptotically stable point under non-steep regularizers, every player plays a pure strategy. The proof combines results presented above. It reaches a contradiction by showing that points in a small enough neighborhood of the asymptotically stable point $x^*$, instead of converging to it as they ought to, they follow recurrent trajectories. In a first step it finds points that after a finite time $T_0$ reach and stay forever at the simplex formed by the support of $x^*$ (using Lemma D.1), which moreover have non-zero volume in that simplex. But then, these points follow FTRL trajectories in the restricted simplex (Proposition B.5) and $x^*$ belongs in the interior of this simplex. However, we already know this cannot be the case in the interior (Theorem C.2) and we follow a similar reasoning.

**Theorem D.2.** *If $x^* \in \mathcal{X}$ is an asymptotically stable point under non-steep regularizers of a generic game $\Gamma$, then it consists of only pure strategies.*

*Proof.* Let $x^* \in \mathcal{X}$ be asymptotically stable, $\mathcal{A}^* = \mathrm{supp}(x^*)$, with $|\mathcal{A}_i^*| = |\mathrm{supp}(x_i^*)| \geq 2$ for some player $i$, and $\mathcal{X}^*$ be its respective simplex. Since $x^*$ is attracting, there exists some (bounded) neighborhood $U$ of $x^*$ for which if $x(0) \in U$, then $\lim_{t \to \infty} x(t) = x^*$. By Lemma D.1, there exists a neighborhood $U_0$ of $x^*$ and a finite time $T_0$ such that if $x(t) = Q(y(t))$ is an orbit of FTRL starting at $x(0) \in U$, then $\mathrm{supp}(x(t)) = \mathcal{A}^*$ for all $t \geq T_0$.

By Proposition B.5, for $t \geq T_0$ all trajectories satisfy Eq. (FTRL-s) and these trajectories coincide with the trajectories of a generic game $\Gamma'$ played on $\mathcal{A}^*$, with the restricted simplex being $\mathcal{X}^*$. At

the same time, similar to the proof of Theorem C.1, $\Phi(U_0, T_0)$ is a bounded (as a subset of $U \cap \mathcal{X}^*$), positive measure for $\mathcal{X}^*$ (as the evolution of an open set after a finite time (recall $|\mathcal{A}_i^*| \geq 2$ for some $i$)), and invariant (Lemma B.6) set of $\mathcal{X}^*$ and, therefore, it is also Poincaré recurrent, contradicting that for $x(0) \in \Phi(U_0, T_0) \subseteq U$, $\lim_{t \to \infty} x(t) = x^*$, as implied by the asymptotic stability of $x^*$. Thus, if $x^* \in \mathcal{X}$ is asymptotically stable then $|\operatorname{supp}(x_i^*)| = 1$ for all $i$. ∎

As we stated in the beginning of this appendix, the steep case of Theorem 4.3 comes as a corollary (Corollary E.2) of a more general result on asymptotically stable sets that we prove in the next section (Theorem 4.5).

# E    Proof of Theorem 4.5 and Theorem 4.3: the steep case

> The following theorem shows that any asymptotically stable set $A$ cannot be contained in the interior of any non-singleton face $\mathcal{X}'$. This comes as a consequence of Theorem C.1. When the regularizers are steep, any point starting in $\operatorname{ri}(\mathcal{X}')$ stays in $\operatorname{ri}(\mathcal{X}')$ over time and $A$ being asymptotically stable implies that $A \cap \mathcal{X}'$ is an asymptotically stable set under the FTRL dynamics of the restricted game played on $\mathcal{X}'$. But for the restricted game Theorem C.1 applies excluding the possibility of $A \cap \mathcal{X}'$ being an asymptotically stable set inside $\operatorname{ri}(\mathcal{X}')$.

**Theorem E.1.** *Let $A \subseteq \mathcal{X}$ be an asymptotically stable set intersecting a non-singleton face $\mathcal{X}'$ of $\mathcal{X}$. Then, $A \cap \mathcal{X}'$ cannot be contained in the relative interior of $\mathcal{X}'$.*

*Proof.* To reach a contradiction let $A$ intersect a non-singleton face $\mathcal{X}'$ of $\mathcal{X}$ and $A' = A \cap \mathcal{X}'$ be a subset of the relative interior of $\mathcal{X}'$. We will show that if this is the case then $A'$ is an asymptotically stable set under the dynamics of FTRL restricted to $\mathcal{X}'$, that lies in the relative interior of $\mathcal{X}'$. This contradicts Theorem C.1.

To reach the contradiction, we go on to prove that $A'$ is an asymptotically stable set under FTRL dynamics in $\mathcal{X}'$. We will crucially use that with steep regularizers for any $x(0) \in \operatorname{ri}(\mathcal{X}')$, $x(t) \in \operatorname{ri}(\mathcal{X}')$, for all $t \geq 0$, i.e., $\mathcal{X}'$ is forward invariant under FTRL (Proposition B.5).

To show Lyapunov stability of $A'$ in $\mathcal{X}'$ pick any neighborhood $U'$ of $A'$ in $\mathcal{X}'$. It can be written as $U' = U \cap \mathcal{X}'$ for some neighborhood $U$ of $A$ in $\mathcal{X}$. Since $A$ is Lyapunov stable, there exists a neighborhood $U_0$ of $A$ in $\mathcal{X}$ such that for any $x(0) \in U_0$, it is $x(t) \in U$ for all $t \geq 0$. Let $U_0' = U_0 \cap \mathcal{X}'$. Using that $\mathcal{X}'$ is forward invariant, the latter implies that for any $x(0) \in U_0' = U_0 \cap \mathcal{X}'$, it is $x(t) \in U \cap \mathcal{X}' = U'$ for all $t \geq 0$, as needed.

We use similar ideas to show that $A'$ is attracting in $\mathcal{X}'$. Since $A$ is attracting in $\mathcal{X}$ there exist a neighborhood $U$ of $A$ in $\mathcal{X}$ such that for any $x(0) \in U$, $x(t) \to A$. Let $U' = U \cap \mathcal{X}'$. The latter combined with the forward invariance of $\mathcal{X}'$ implies that for any $x(0) \in U \cap \mathcal{X}' = U'$, $x(t) \to A \cap \mathcal{X}' = A'$, as needed. ∎

> As a corollary of the above theorem we may get Theorem 4.5, restated below. Whenever an asymptotically stable set intersects a non-singleton face it must intersect its respective boundary, and thus a face of smaller dimension. Consequently, "in the long run", it must intersect a singleton face. Put differently, it should contain a point consisting of only pure strategies.

**Theorem 4.5.** *Every asymptotically stable set of steep (FTRL) contains a pure strategy.*

*Proof.* Let $A$ be asymptotically stable and $\mathcal{X}_{min}$ be a face of minimal dimension intersected by $A$ which is not a singleton. By Theorem E.1, $A$ cannot be contained in the relative interior of $\mathcal{X}_{min}$, so it must intersect the boundary of $\mathcal{X}_{min}$. However, this means that $A$ intersects a face of dimension strictly smaller than that of $\mathcal{X}_{min}$, a contradiction. Thus, $\mathcal{X}_{min}$ is a singleton and $A$ must contain a vertex of $\mathcal{X}$. ∎

**Corollary E.2.** *If $x$ is an asymptotically stable point, then it consists of only pure strategies.*