[Reviews · NeurIPS 2020]

Review 1

Summary and Contributions: The paper studies multiplayer genaral-sum games where each player has a finite set of actions. The main finding of this paper is that if asymptotically stable points of FTRL must be pure Nash equilibria. Equivalently, mixed Nash are not stable under FTRL.

Strengths: The relevance of the results, analysis and significance are quite impressive. The paper is well written and the problem is well placed into context.

Weaknesses: It may be a good idea to add some comments in the Broader Impact section regarding the practical implications of this result.

Correctness: Claims seem to be correct.

Clarity: yes.

Relation to Prior Work: yes

Reproducibility: Yes

Additional Feedback: Line 48: typo overcoe -> overcome Paragraph 282: I get the feeling that you are not happy with the definition of Nash equilibria because regret minimization by itself is not guaranteed to find it. Don't you think that the problem could instead be that individual regret minimization is simply not good enough? I have a few general and somewhat off topic questions. -Do you have any intuition on what could happen to the average iterate in general games? For example MW cycles around in zero sum matrix games but it is known that the average iterate is close to a Nash. -It is well known that in some games, no-regret algorithms do not imply convergence to Nash. Hence why we are looking for last iterate convergence in algorithms see (Abernethy, Jacob, Kevin A. Lai, and Andre Wibisono. "Last-iterate convergence rates for min-max optimization." arXiv preprint arXiv:1906.02027 (2019)) and references within. What do you think are good next steps for actually finding NE, averaging? optimism? UPDATE AFTER AUTHOR FEEDBACK: Thanks for the clarifications! I think this is good work so will leave my score unchanged.


Review 2

Summary and Contributions: This paper studies the convergence properties of follow the regularized leader (FTRL) in general N-player normal-form games. While it is well-understood that the empirical frequency of play generated by FTRL (as well as any other no-regret algorithm) approaches the set of coarse correlated equilibria, its relation with Nash equilibria is still largely unexplored (except for some results on zero-sum games). The paper fills this gap by showing that mixed Nash equilibria cannot be asymptotically stable points of the dynamics generated by FTRL, as only pure Nash equilibria can enjoy this property. The key fact used to prove this result is that the dynamics of FTRL preserve volume in the space of the players’ payoffs, which allows to immediately get the result when FTRL is run using regularizers that allow to reach the borders of the strategy space only in the limit (such as entropic regularization). Instead, for regularizers that do not satisfy this property, such as L^2 regularization, the same result is proven with a much more intricate analysis, as the main difficulty is that the strategy profiles generated by the FTRL dynamics might change support in finite time (instead of always having full support in finite time).

Strengths: I believe that the convergence analysis of no-regret learning algorithms in games is a topic of great interest nowadays, not only for the algorithmic game theory community, but also for a broader audience interested in general online learning. The results presented in the paper considerably advance the state of the art, showing that FTRL is antithetical to mixed Nash equilibria, and that only pure Nash equilibria can be obtained in this way. This opens new research problems, such as similar converge analysis for other no-regret learning algorithms. The techniques used to prove the results are non-trivial as they exploit advanced mathematical tools form the theory of dynamical systems.

Weaknesses: A minor weakness of the paper is that all the results only hold for normal-form games. Given the pervasiveness of no-regret learning in sequential games, I would have expected at least some discussion on which results carry over to this more general setting (see questions below).

Correctness: As far as I am concerned, all the results presented in the paper are sound, even though I did not carefully check all the technical proofs in the supplemental material.

Clarity: Overall, the paper is well written, though some adjustments may improve readability and resolve typos: Line 48: “overcoe” -> “overcome”. Lines 194-204: This paragraphs are too dense and hard to follow. Line 213: Double “in”. Lines 227-271: Use \itemize to fix spacing and indentation. Line 246, at the end: Should it be U instead of U’? Lines 273-281: This paragraphs is too dense and hard to follow. Line 308: “of \X^\ast of \X”, it is not clear.

Relation to Prior Work: The relations with the state of the art is adequately commented.

Reproducibility: Yes

Additional Feedback: I have a question to the authors. Do you have any clues on how your results carry over to more general game settings such as extensive-form games? AFTER AUTHOR FEEDBACK The authors answer my curiosity. They should add one sentence in the paper about the extension to extensive-form games.


Review 3

Summary and Contributions: This paper provides a theoretical analysis of the convergence behavior of follow the regularized leader (FTRL), a widely used no-regret learning algorithm, and proves that except under special circumstances it does not converge to a Nash equilibrium.

Strengths: The results from this paper seem highly impactful. The paper is also well-written given the complex subject and contains a good overview of previous work.

Weaknesses: I do not see any clear weaknesses in this paper

Correctness: The theory in the paper requires I background that I unfortunately lack, so I cannot comment on the correctness of the proofs. To the extent that I can tell, it does seem to be correct.

Clarity: The paper is well written. Although I could not understand all of the theory in the paper, I could still understand the paper at a high level.

Relation to Prior Work: There is a good discussion of prior work and how this paper differs from them. I am not familiar with the prior work in this area so I cannot say how novel these results are compared to prior work.

Reproducibility: Yes

Additional Feedback: On line 104, the authors point to prior related work that was focused only on zero-sum games. But any N player general-sum game can be converted to a N+1 player zero-sum game. So why does that result not extend to general-sum games? The authors very briefly specify on line 153 that the regret being discussed in this paper is external regret. It might be worth expanding on this a bit and pointing out that other forms of regret exist that result in convergence to other forms of equilibria, such as correlated equilibrium. In a few places, such as line 179, the authors say that they will not dive into the details of certain background material. That's fine, but given that these are short papers it might be worth just removing such comments to save space. UPDATE AFTER AUTHOR FEEDBACK: Thank you for answering my question about the zero-sum result. I think it should be specified in the paper that those results apply only to polymatrix zero-sum games.


Review 4

Summary and Contributions: ############################################################ I read the authors' response. I did not have any major concerns to begin with so I will keep my score. ############################################################ The present work is concerned with follow-the-regularized-leader (FTRL) dynamics in the setting of general sum multi-agent games with finite decision space. It proves that for a Nash equilibrium to be attractive under FTRL dynamics using either squared normor the (negative) Shannon Entropy as a regularizer, said Nash equilibrium has to be strict, meaning that each player has a unique best response given that the other player's are playing the Nash equilibrium. This proof relies on the insight that FTRL dynamics in the space of pay-offs are volume preserving. While this observation has been made before, in special cases, but the authors extend it to show that any game, under any FTRL dynamics has this property. The authors use this insight to elegantly prove their result for the squared norm and the shannon entropy regularization, observing that they form archetypical examples of a dichotomy between steep (dynamics restricted to interior of probability simplex) and non-steep regularizers.

Strengths: This is really nice work! It provides a mathematically elegant proof of a fundamental result in online learning. Since it is well-written and contains many interesting techniques, I am optimistic that it can have impact beyond its main result, as well.

Weaknesses: I don't have any major concerns.

Correctness: I did not check the derivation in the appendix in detail, but the results and proof strategy were plausible.

Clarity: The paper is very well written, with plenty of illustrations that ease understanding.

Relation to Prior Work: Prior work is clearly discussed, although I am not too familiar with the literature on online learning

Reproducibility: Yes

Additional Feedback: It might be worth considering to also advertise the result on incompressibility of the payoff dynamics in the abstract.

[Author Response · NeurIPS 2020]

We would first like to thank the reviewers for their insightful comments on our work. We appreciate that you like our paper and that you are helping us making it even stronger with your comments. We will happily incorporate all of your suggestions for improving our paper.

**Opinion on Nash equilibrium.** Reviewer #1 felt that we were critical of the notion of NE based on line 282. We definitely did not intend to give the impression that we are "unhappy" with the notion of Nash equilibrium: it is a natural and elegant solution concept, and we believe its role as a "gold standard" in game theory is well-deserved. Line 282 concerns the stability of **mixed** Nash equilibria: in contrast to *strict* equilibria, mixed equilibria are fragile because a small "trembling hand" imperfection in the agents' mixed strategies could lead to off-equilibrium behavior. They also effectively presuppose coordination between the agents as, in a mixed equilibrium, any agent could deviate to another mixed strategy with the same support without any penalty; however, doing so would destroy the equilibrium. These shortcomings of mixed equilibria are well documented in the game theory literature. By contrast, *strict* equilibria are uniquely robust in this regard (since they are preserved by small payoff perturbations), and we find the fact that this robustness is picked up by no-regret strategies (like FTRL) quite important and insightful from a learning standpoint. In a certain sense, this provides a crucial link between game-theoretic learning and the extensive equilibrium refinement literature in game theory.

**Effects of averaging in general games.** Reviewer #1 asked whether averaging can help in general games. There exist small, simple games for which even the time-average of regret-minimizing dynamics does not converge to Nash equilibrium (either in terms of strategies or payoffs) [1, 3]. In fact, [3] shows an example of a class of games with no pure NE such that for all but a zero-measure of initial conditions replicator converges to a limit cycle whose social welfare everywhere (and hence its time average) is optimal and can be arbitrarily better than the best Nash. Thus, time averaging cannot hope to restore any meaningful connection between regret minimization and NE in general games.

**Discovering Nash equilibria.** Reviewer #1 inquired about promising approaches to finding NE. Optimism can help resolve very special cases of games where the payoff field is monotone without being strictly/strongly monotone (e.g., in bilinear zero-sum games with an interior equilibrium). One can construct game instances where optimistic approaches do not necessarily converge to meaningful equilibria, e.g., [2]. Given that optimism is a much newer technique its failure modes are less well understood, but we suspect that similar to averaging even stronger negative results are possible. Given the PPAD completeness of the problem, there is little hope that a non-exhaustive method can help in the worst case.

**Extensive form games.** Reviewer #2 wanted to know which of our results carry over to extensive form games. That is an excellent and important question! A key component of our proof is that the FTRL dynamics are divergence free in the payoff space for normal form games. For sequential imperfect information game setting under perfect recall FTRL remains divergence free [5]. It should be straightforward to extend our results for these cases, modulo a heavy notational overhead. Although this direction lies outside our current scope, it should be clear that the insights of our paper are valuable for this line of work and we hope that they will trigger more theoretical/experimental investigations.

**Lifting to $(N + 1)$-player zero-sum games.** Reviewer #3 wants to know if our results can be directly derived from the zero-sum case of [4] by lifting $N$-player games to $(N + 1)$-player zero-sum games. [4] captures only *polymatrix* zero-sum games, i.e., games that can be described by a network of zero-sum *bimatrix* games. General $N$-player games cannot be lifted to $(N + 1)$-player polymatrix zero-sum games. This distinction is important because otherwise all FTRL dynamics on games with an interior Nash equilibrium would have recurrent trajectories based on [4].

# References

[1] C. Daskalakis, R. M. Frongillo, C. H. Papadimitriou, G. Pierrakos, and G. Valiant. On learning algorithms for nash equilibria. In *Symposium on Algorithmic Game Theory 2010*, 2010.

[2] C. Daskalakis and I. Panageas. The limit points of (optimistic) gradient descent in min-max optimization. In *NeurIPS 2018*, 2018.

[3] R. D. Kleinberg, K. Ligett, G. Piliouras, and É. Tardos. Beyond the nash equilibrium barrier. In *Innovations in Computer Science 2011*, 2011.

[4] P. Mertikopoulos, C. H. Papadimitriou, and G. Piliouras. Cycles in adversarial regularized learning. In *Symposium on Discrete Algorithms 2018*, 2018.

[5] J. Pérolat, R. Munos, J. Lespiau, S. Omidshafiei, M. Rowland, P. A. Ortega, N. Burch, T. W. Anthony, D. Balduzzi, B. D. Vylder, G. Piliouras, M. Lanctot, and K. Tuyls. From poincaré recurrence to convergence in imperfect information games: Finding equilibrium via regularization. *CoRR*, abs/2002.08456, 2020.


[Meta-Review · NeurIPS 2020]

Your paper was carefully reviewed, and all four reviewers were quite impressed by the work. The results are significant, particularly the characterization of when exactly FTRL dynamics converges to NE. Also the writing is reasonably strong. Please keep in mind the review comments when compiling your paper for the camera-ready version.